# Hypoxia in Cancer and Fibrosis: Part of the Problem and Part of the Solution

**DOI:** 10.3390/ijms22158335

**Published:** 2021-08-03

**Authors:** Yair Romero, Arnoldo Aquino-Gálvez

**Affiliations:** 1Facultad de Ciencias, Universidad Nacional Autónoma de México, Mexico City 04510, Mexico; 2Instituto Nacional de Enfermedades Respiratorias “Ismael Cosío Villegas”, Mexico City 14080, Mexico

**Keywords:** hypoxia-inducible factors, lung cancer, idiopathic pulmonary fibrosis, regeneration

## Abstract

Adaptive responses to hypoxia are involved in the progression of lung cancer and pulmonary fibrosis. However, it has not been pointed out that hypoxia may be the link between these diseases. As tumors or scars expand, a lack of oxygen results in the activation of the hypoxia response, promoting cell survival even during chronic conditions. The role of hypoxia-inducible factors (HIFs) as master regulators of this adaptation is crucial in both lung cancer and idiopathic pulmonary fibrosis, which have shown the active transcriptional signature of this pathway. Emerging evidence suggests that interconnected feedback loops such as metabolic changes, fibroblast differentiation or extracellular matrix remodeling contribute to HIF overactivation, making it an irreversible phenomenon. This review will focus on the role of HIF signaling and its possible overlapping in order to identify new opportunities in therapy and regeneration.

## 1. Hypoxia Adaptation and Signaling

The capacity to detect oxygen levels is essential for the survival of organisms and cells, especially within the lung. When the cell has enough oxygen (normoxia), alpha subunits are degraded by the ubiquitin–proteasome pathway with the intervention of Von Hippel Lindau protein (VHL). This interaction with VHL depends on available oxygen as the substrate of hydroxylation reactions by prolyl hydroxylases (PHD1-3) [1,2,3,4]. Instead, when the oxygen concentration declines, these reactions are inhibited, which causes alpha subunit accumulation in the cytoplasm and subsequent translocation to the nucleus, where it heterodimerizes with the β subunit [5]. These heterodimers within the nucleus recognize the sequence TACGTG, called the hypoxia response element (HRE), located in the promoter regions of genes involved in hematopoiesis, angiogenesis, iron transport, glucose consumption, MEC synthesis, cell growth and differentiation, among other functions [6].

HIF-1α was identified by examining the regulatory mechanism of erythropoietin thirty years ago [7,8,9]. Later, HIF-2α was recognized by comparing the sequence with HIF-1α and it was found with the endothelium (EPAS1) [10,11]. The third member of this family was identified in 1998 [12]. The three isoforms share heterodimers with a beta subunit constitutively expressed, called HIF-1β (also known as ARNT); see Figure 1. 

### Structure and Functional Domains of Hypoxia-Inducible Factors

The structure of these proteins consists of different domains: Basic helix–loop–helix (bHLH). This is a basic region followed by two alpha-helices separated by a variable loop region, and this region confers the ability to bind DNA as homodimers or heterodimers, which is found in other transcription factors [13].PAS domains. PAS-A and PAS-B domains also build a stable heterodimer that enables robust DNA binding. In the PAS-B structure, a central β-sheet provides a hydrophobic surface for HIF-1β that is conserved among the HIFα isoforms [14]. Oxygen-dependent degradation domain (ODDD). This domain is responsible for degradation by the ubiquitin–proteome system, located ~200 amino acids in the central region and has two functional domains: N-terminal (NODD) and the C-terminal (CODD) [15]. The hydroxylation of NODD is more sensitive to hypoxia than CODD; thus, it is possible to block them differentially [16]. The transactivation domain (TAD). These domains (N-TAD and C-TAD) contain binding sites for other proteins such as transcription coregulators essential in determining what type of genes will be expressed due to specific recruitment [17,18]. Nuclear localization signal (NLS). α and β subunits have two nuclear localization sequences in N- and C-terminal domains [19]. Direct interaction of HIFs with importins 1α, 3α, 5α and 7α has been described, and this interaction depends on a nuclear localization signal within the C-terminal region [20]. Furthermore, an additional interaction site has been reported with importins 4 and 7, which is more efficient than NLS interaction [21]. The nuclear export signal (NES), located in amino acids 616–658, has been reported to be regulated by MAPK phosphorylation [22].

## 2. Cancer Hallmarks Associated with Hypoxia 

Lung cancer and idiopathic pulmonary fibrosis (IPF) compromise the lung parenchyma, with an irreversible loss of gas exchange; solid tumors and fibroblast foci have limited nutrient supply due to the dense microenvironment; counterintuitively, cells still proliferate, causing disease progression. In this context, hypoxia adaptation could represent a coherent link between cell proliferation and the interruption of the nutrient supply [23]. Although the relationship between cancer and fibrosis has been reported [24,25,26,27], hypoxia has not been pointed out as the cause of this association. Nevertheless, there is a compelling body of literature about hypoxia in both diseases. Therefore, it is necessary to identify the elements described in the cancer literature and the potential overlap.

### 2.1. Hypoxic Microenvironment in Tumors

Initially, the role of hypoxia in cancer was restricted to the “tumor microenvironment” of solid tumors. Tumors have necrotic areas close to areas with gradients of oxygen related to the distance to the blood vessels. Although these gradients are also present in normal tissues, in cancer, these gradients decline abruptly and chronically [28,29]. Furthermore, the cyclic hypoxia offers a strong activation of the HIF pathway, which, in turn, helps tumor cells to survive adverse conditions such as cytotoxic therapy [30,31]. Dewhirst et al. explain this overactivation with an analogy about the frequency of the cycles of tides and waves and how these impact on the coast, where tides represent the degree of hypoxia and waves the high-frequency fluctuations of the blood flux. When there is an overlap, overactivation of HIF pathway occurs [30,32]. An important example of this association between cancer and the cyclic hypoxia might be appreciated in cancer patients suffering from obstructive sleep apnea (OSA); this association participates in increasing the incidence and mortality of cancer [33]. Several studies have confirmed that exposure to cyclic hypoxia increases cancer stem cell markers, as well as features related to metastasis [34,35,36]. It has even been suggested that the aggressiveness in melanoma can be mediated by TGF-β (a well-known pro-fibrotic cytokine) [37]. This association is also present in other diseases, such as hypertension and type 2 diabetes, in which there is a particular adaptation of HIF signaling resulting in the overactivation of HIF-1α and downregulation of HIF-2α and a consequent increase in reactive oxygen species (ROS) [38,39]. ROS are an integral component of the tumor microenvironment as they engage HIF stabilization and immunity regulation [40,41]. Thus, the kinetics and integration of the response to hypoxia in the tumor microenvironment are complex.

### 2.2. Hypoxia-Induced Metabolic Changes

In addition to these hypoxic areas, another widely studied mechanism is the metabolic reprogramming of cancer cells, described almost 100 years ago by Otto Warburg [42]. From initial observations about increased glycolysis in tumors even in the presence of oxygen, this phenomenon has been implicated with the intense proliferation of cancer cells [43,44]. The molecular basis of this adaptation is similar to the fermentation in hypoxic conditions, which was clarified by the understanding of HIF signaling and the role of mitochondria in its activation [40]. This bioenergetic switch induces the expression of multiple genes involved in glycolysis; although this phenomenon goes beyond a change in glucose metabolism, this implies integration with biosynthetic pathways [45,46,47]. For example, other intermediates of the tricarboxylic acid cycle (TCA) are used in macromolecule synthesis [47]. In the same line, cancer cells in general show an addiction to glutamine; its role transitions from being a donor in the biosynthesis of nucleotides and amino acids to the activation of the mTOR complex (which will be discussed later) [48]. With this knowledge, glutaminolysis has been incorporated into metabolic cancer treatment [49,50,51].

### 2.3. Integration of HIFs Signaling in Lung Cancer

Despite the considerable growth of the hypoxia field, our understanding remains restricted in how HIF circuitries are integrated and especially by the fact that distinct alpha subunits can elicit different responses. HIF-1α has been reported in a large number of cancer types; its primary association is in an acute response with the expression of glucose transporters, glycolytic enzymes and growth factors [52]. HIF-2α plays a predominant role in chronic hypoxia and angiogenesis [53]. This could be partially explained by the fact that HIF-2α is relatively more resistant to factor-inhibiting HIF1 (FIH-1) [54]. Although there is a certain redundancy between HIF1 and 2, HIF-2α has a predilection for MMP14, EPO and PAI-1 [55,56,57]. In addition, a loss of HIF-2α causes a fatal respiratory distress syndrome in mice since HIF-2α regulates the activation of VEGF in lung maturation and surfactant production by type 2 alveolar cells [58]. In the context of lung cancer, HIF-1α and HIF-2α are overexpressed in both the cytoplasm and the nucleus and are associated with poor prognosis [59]. The HIF-3α locus allows extensive alternative splicing, producing several variants; this variety confers HIF-3α a broader and complex role [60]. For instance, HIF-3α1 inhibits transcriptional activation by the lack of a transactivation domain (TAD), while HIF-3α4 inhibits nuclear translocation [61]. However, knock-down of HIF-3α induces downregulation of the HIF signature [61]. Recently, we found that treatment with a combination of 2-methoxyestradiol and sodium dichloroacetate in A549 cells reduces hypoxia-induced resistance at 1% O_2_ for 72 h, mainly by the effect in HIF-3α and a modest contribution of HIF-1α [62]. This exemplifies how hypoxia signaling is orchestrated by distinct members on the pathway.

An essential element in this adaptation is that hypoxia can directly alter the epigenetic landscape. It has been reported, in cancer and aging, that there is a global loss of methylation marks along with hypermethylation in specific promoters [63,64]. For example, the hypoxic condition affects DNA demethylation by reducing TET enzyme activity [65]. Furthermore, hypoxia-induced chromatin remodeling might be mediated by histone deacetylases (HDACs) and histone demethylase enzymes, with a Jumanji-C (JmjC) domain [66,67]. Definitively, epigenetics denotes the possibility of the integration of microenvironment signals with metabolism. Thus, hypoxia adaptation by HIF signaling (mainly overactivation of HIF-1α) promotes aggressiveness by several mechanisms in cancer cells but without establishing if there is a cause of chronic exposure [52,68].

### 2.4. Interconnected Pathways with Hypoxia

The molecular sensors involved in stress responses have feedback mechanisms that are necessary to adapt to the microenvironment. An example of these interconnections is the link between mTOR and hypoxia. mTOR complex kinase lies at the nexus of many signaling pathways involved in cell growth by protein synthesis and inhibition of autophagy [69]. Although both are master regulators of metabolism, it is difficult to understand that mTOR is inhibited by hypoxia conditions (typically in “normal cells”), and hypoxia-inducible factors’ synthesis is mTOR-dependent. This makes sense in a gradual synchronic response: at the beginning, moderate hypoxia stress may act as an obstacle for survival and proliferation, but, later, severe hypoxia could drive the assortment of mutations that confer resistance to cell death, such as TP53 mutations [70,71]. It is important, in this first scenario, to consider the possibility to reduce autophagy in order to eliminate a probable route of nutrient supply [72]. Otherwise, mTOR and the unfolding protein response (UPR) contribute to hypoxia tolerance [73,74,75]. In this chronic exposure, the participation of hypoxia in tumorigenesis is significant; for example, in patients with non-small-cell lung cancer, recurrent mutations of the epidermal growth factor receptor (EGFR) have been associated with hypoxia [76,77]. Thus, hypoxia pathway overactivation corresponds to an activation of alternative redundant pathways involved in cell survival [3,78]. Moreover, it is essential to note that the lack of oxygen is not the only mechanism behind HIF activation. Additionally, this effect can be observed by oncogenes or mutations in tumor suppressors such as PTEN, VHL and p53 or directly by redox imbalance [79,80]. Moreover, this activation of HIFs, either oxygen-dependent or independent, can be found in the same tumor [81]. 

### 2.5. Hypoxia and Extracellular Matrix Remodeling

Generally, cancer progression is accompanied by dense accumulations of the extracellular matrix that supports the structure of the tumor (stroma). The stroma is synthesized by cancer-associated fibroblasts (CAF); these cells are phenotypically different from normal fibroblasts and acquire markers such as alpha smooth muscle actin (αSMA), fibroblast activation protein (FAP), α1β1 integrin and Thy-1 [82]. Notwithstanding, this phenomenon makes the overlapping between cancer and fibrosis clear as there are some contradictions about the role of these fibroblasts in cancer. The different CAF subpopulations depend on their origin and on the signals in the microenvironment [83]. At present, several origins have been described, such as epithelial to mesenchymal transition (EMT), activation of resident fibroblasts, endothelial to mesenchymal transition (EndMT) and circulating bone marrow-derived cells [84]. In terms of the microenvironment, stiffness of the extracellular matrix or hypoxia drive the activation of CAF, where feedback loops that support the progression of cancer have been established. Several reports are pointing out the role of MMPs and specific growth factors (mainly TGF-β) in the remodeling of the extracellular matrix (ECM); currently, nevertheless, it is becoming clear how this tissue remodeling integrates mechanical forces. For example, the ECM produced by CAFs forms a capsule adjacent to cancer cells; this capsule actively compresses cancer cells by actomyosin contractility forces [85]. Furthermore, the tumor is broader than epithelial cells and CAFs; there also exists a complex interaction of distinct cell populations (cancer stem cells, endothelial cells, inflammatory cells, pericytes) and the clonal heterogeneity that could be explained (at least in part) by hypoxia gradients [28]. 

In summary, from this relation between cancer and hypoxia, we can extract concepts such as a hypoxic microenvironment in tumors, cyclic hypoxia, metabolic changes, extracellular matrix remodeling and interconnections that can be extended to fibrosis and vice versa (Table 1). 

## 3. Hypoxia Signaling Overlap in Cancer and Fibrosis?

Historically, it has been described that idiopathic pulmonary fibrosis increases the risk of developing cancer. IPF increases the risk to 20%, and the fibrotic lesion areas have the highest incidence of tumors [25,88]. However, the link to this association is unclear. In general, the cause has changed over time; for example, the first association was smoking habits, which was disregarded [89]. Later, inflammation was proposed as a cause; nowadays, it does not have a major role in the pathogenesis of IPF [90]. Here, we postulate that this association, at least in part, is due to hypoxia with nodes that participate in progression (Figure 2).

### 3.1. Hypoxia in the Pathophysiology of Idiopathic Pulmonary Fibrosis

In brief, the pathophysiology of IPF is caused initially by an aberrant response of the alveolar epithelium [90,91]. This aberrant response is forced by a stochastic profibrotic age-related epigenetic drift [92,93,94]. These cells are found in close relationship with active sites of fibrogenesis sites (fibroblast foci) and are responsible for the secretion of many mediators for fibroblast recruitment/activation, such as PDFG, TGF-β, MMP7, OPN, endothelin-1 and more [95]. It is important to note that these markers have also been associated with hypoxia in cancer [26,96]. For example, MMP7 and MMP14 (also known as MT1-MMP) participate in metastasis, and their increase is associated with a hypoxic microenvironment [86]. In IPF, MMP7 was described as a profibrotic biomarker, and its expression is in a relationship with OPN increase, which is also associated with cancer and hypoxia [87,97,98,99]. In the case of MMP14, we recently published findings indicating that it protects against fibrosis by preventing the senescence of alveolar type 2 cells [100]. Although its expression is dependent on HIF-2α, it has not been established in IPF [55]. Interestingly, HIF-2α and MMP14 participate in the embryonic development of lung through the integration of pulmonary vasculature and alveoli; knockout mice for these proteins confirm that the lethality is due to inadequate alveolarization [58,101]. This process is also recapitulated in regeneration and will be discussed later.

Additionally, the role of other cell populations (in addition to epithelial cells and fibroblasts) has been little studied in IPF. However, it has been reported that there is a population of macrophages that can promote fibrosis through immunological modulation [102]. During early events of fibrosis, macrophages have been implicated in the activation of host defense and profibrotic mechanisms [103]. Furthermore, a hypoxic microenvironment has an altered metabolite composition that promotes the activation of immune cells [104]. Accordingly, the link might be through elevated mitochondrial ROS or TCA cycle metabolites already involved in the regulation of immune cells [104,105]. 

### 3.2. Effects of Hypoxia on the Alveolar Epithelium

Further effects of hypoxia on alveolar epithelial cells include increased surfactant production, disruption of cytoskeleton integrity and apoptosis [106,107,108]. The adaptation mechanisms of these cells depend on the severity as well as on the duration of hypoxia, from adjusting ATP consumption in short periods to epithelial to mesenchymal transition (EMT) or cell death in chronic exposure [108]. One speculative feedback between hypoxia and fibrosis is that hypoxic conditions could provoke an impairment of surfactant activity and the abnormal remodeling of the alveolar space, suggested as an early marker of fibrosis, associated with aging [109,110,111,112]. Moreover, after 48 h of hypoxia, alveolar epithelial cells type 2 (AEC2) activate apoptosis, which promotes fibrosis [107,113]. In addition, surviving epithelial cells are an additional source of fibroblasts through the EMT phenomenon, which has been consolidated in the pathophysiological basis of cancer and fibrosis [26,27,67].

### 3.3. Profibrotic Feedback Loops Related to Hypoxia

Even though epithelial cells play a leading role in the onset of the disease, the IPF progression depends on the establishment of profibrotic feedback loops, where hypoxia is implicated. The basis of this notion is that exacerbated extracellular matrix deposition in fibroblast foci promotes hypoxia and vice versa (Figure 3). Again, the microenvironment in the fibroblast foci of IPF determines fibroblast activation by the relation between the stiffness of the ECM and the degree of hypoxia. Evidence indicates that IPF lungs show transcriptional activation of hypoxia (mainly HIF-1α), and the knockout mice for HIF-1α showed reduced lung fibrosis [114,115]. Furthermore, aggressive phenotypic changes in fibroblasts can be attributed to hypoxia. For example, loss of Thy-1 has been associated with a profibrotic phenotype that favors proliferation, migration, ECM secretion and TGF-β signaling [116]. This loss has been reported to be due to hypoxia-induced hypermethylation by hypoxia [117]. In cancer, Thy-1 has been proposed as a tumor suppressor, suggesting an overlapping role in both diseases [118,119]. In addition, in an integral overview (measuring the three alpha subunits), IPF fibroblasts showed specific hypermethylation of HIF-3α, which indicates that the lack of HIF-3α causes the overactivation of hypoxia signaling and increases differentiation to myofibroblasts [120]. Thus, IPF fibroblasts have epigenetic changes, perhaps associated with aging, which cause their overactivation [121]. Furthermore, it has been suggested that fibroblasts form a reticulum that infiltrates the lung parenchyma, similar to a neoplasm [122,123].

In a similar way to cancer, cyclic hypoxia increases the aggressiveness of fibrosis in patients with obstructive sleep apnea (OSA). For example, a study in the bleomycin-induced fibrosis model shows that cyclic hypoxia by redox imbalance increases the severity of the lesions and amplifies the collagen deposition, with a consequent increase in ECM stiffness [124]. Although the molecular mechanism of this deterioration remains unknown, the possibility of shared common pathogenetic pathways suggests that the management of these “comorbidities” is equally crucial in IPF treatment [125,126]. 

In addition, fibroblast foci have characteristics expected in a hypoxic environment, such as anaerobic metabolism. Although metabolic change is a field modestly addressed in idiopathic pulmonary fibrosis, it is involved in the survival of cells against nutrient stress, differentiation to myofibroblasts and collagen production [106,127,128]. Lactic acid, from anaerobic metabolism, has been proposed as an important mediator of myofibroblast differentiation through the pH-dependent activation of TGF-β [129,130]. Myofibroblasts from IPF have increased glycolysis and increased differentiation, mediated by HIF-1α and TGF-β [131]. It has also been shown that during aging, an increase of GLUT1 promotes fibrogenesis [132]. In summary, these metabolic changes mediate a transition from oxidative to glycolytic metabolism, similar to cancer [133,134].

Therefore, hypoxia signaling is overactivated in lung tumors and pulmonary fibrotic injuries, with profound alterations of several pathways that are interconnected (Table 1 and Table 2). In this context, hypoxia supports disease progression mainly by its role in the survival and the establishment of profibrotic loops. However, the potential role in regeneration is exciting. 

## 4. Nothing in Hypoxia Signaling Makes Sense except in Light of Regeneration and Development

The adult lung is a largely quiescent tissue; the alveolar epithelium has a turnover of only 7% of alveoli per year [140]. This epithelium is responsible for gas exchange and is composed exclusively of alveolar epithelial cells type 1 and 2. The challenge in respiratory diseases is the continuous identification of nodes that lead to a pathological state such as cancer or fibrosis [141]. Consequently, it will be significant to investigate how epithelial cells can respond effectively to hypoxia injury by activating stem/progenitor populations. In the adult lung, the interaction of alveolar epithelial cells type 2 and mesenchymal cells (fibroblasts) is responsible for recovering the alveolar epithelium when an injury occurs [142,143]. In this sense, these cells, as facultative progenitors, require a microenvironment or “niche” that detonates their function [144]. Several studies reported that hypoxia and HIFs have a crucial role in promoting an undifferentiated state by the interaction with their respective niches by OCT4 or Sox2, or signaling pathways such as Notch and Wnt [135,145,146,147]. One example of these niches can be found in the bone marrow, where hematopoietic stem cells reside in regions with low oxygen tension [148]. In lung development, this phenomenon also occurs, where oxygen availability determines the patterning and morphogenesis of the trachea, as observed in studies of D. melanogaster [149,150]. In addition, HIF signaling regulates alveologenesis during development, predominantly by HIF-2α and HIF-3α [58,151]. 

Hypoxia overactivation suggests that regeneration has been initiated but not completed, which is due to recurrent injury or inadequate regeneration (Figure 3). For example, during lung regeneration, hypoxia develops a level of response that determines epithelial fate; after influenza infection, these lesions have an acute activation, while in IPF cells, hypoxia is severe, with changes in the behavior of epithelial cells, such as lower surfactant production [136]. Perturbations associated with aging, mutations and epigenetic modifications could remove the breaks on hypoxia signaling in cancer and fibrosis, which could be the cause of these aberrant persistence of stem cell states [152]. For example, in fibrosis, where cells attempt to restore the lung, the HIF pathway signature is also present in the stem cell states [137,153]. In the same line, in IPF, a recent paper highlights a particular population of basaloid cells in the surface of fibroblasts foci that coexpress epithelial, mesenchymal, senescence and developmental markers [154]. Further transition states have been reported due to mechanical forces or spontaneous forms that need to be incorporated [155,156]. Understanding lung regeneration is a fundamental challenge for future therapies.

Finally, integrating this perspective will offer the possibility of new therapies. For example, mesenchymal stem cells cultured in hypoxia reach senescence later [138]. In the same cells, hypoxia pretreatment improves their transplantation in a fibrosis model [139]. It is important to note that the relationship between cancer and fibrosis makes clear the possibility of using the same treatments, such as the current IPF therapies, pirfenidone and nintedanib, that are also active in lung cancer [25]. Furthermore, there is the possibility of using drugs activated by enzymatic reduction in hypoxic tissue [157].

In summary, this review provides the basis (along with other papers) to integrate the hypoxia response from the molecular level with the intercommunicated pathways that form some of the circuitries in respiratory diseases.

## 5. Conclusions

Lung cancer and pulmonary fibrosis (particularly IPF) are chronic diseases associated with hypoxia and represent a significant public health problem. This review makes clear the relationship that exists between cancer and fibrosis through hypoxia, not only in a transcriptionally active signature but also in feedback loops, which may be the basis for progression in these two diseases. Although the hypoxia response is necessary for the regeneration process, if it persists, it may lead to activation of feedback loops related to progression, which generally overlap between these diseases. To face this challenge in cancer and fibrosis, an integrated understanding of the crosstalk between the development and regeneration of stem cells with the external environment is required.

## Figures and Tables

**Figure 1 ijms-22-08335-f001:**
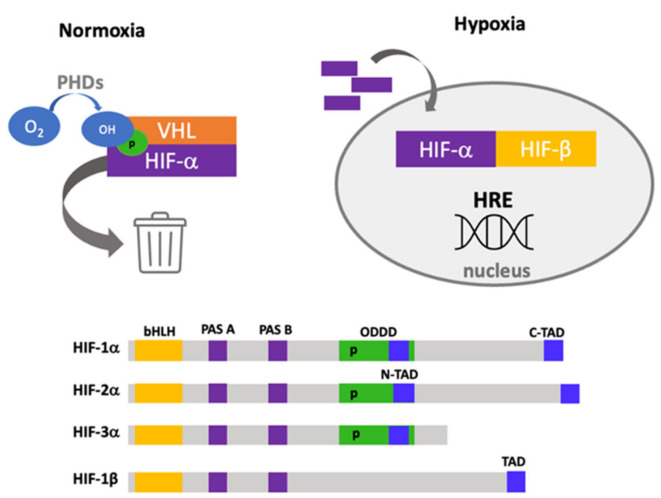
Hypoxia adaptation by HIF system. Hydroxylation reactions in proline residues of alpha subunits (HIF-1α, HIF-2α and HIF-3α) depend on available oxygen (normoxia); these reactions cause its degradation by interaction with Von Hippel Lindau protein (VHL). The decrease in oxygen concentration inhibits HIF-α hydroxylation and induces its accumulation in the cytoplasm and subsequent translocation to the nucleus, where it heterodimerizes with the HIF-1β (hypoxia). This heterodimer is able to bind to hypoxia response elements (HRE) found in promoters of diverse genes. Structure and functional domains of hypoxia-inducible factors (below).

**Figure 2 ijms-22-08335-f002:**
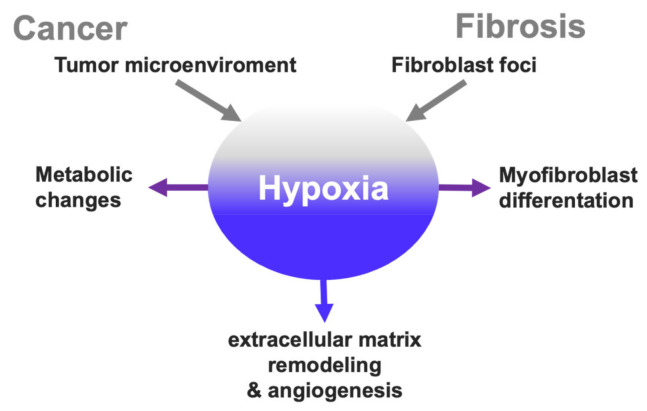
Hypoxia drives the progression in cancer and fibrosis. Hypoxia is an important part in the integration of the microenvironment signals; both tumors and fibroblast foci induce an activation of hypoxia and this in turn activates different mechanisms involved in its progression.

**Figure 3 ijms-22-08335-f003:**
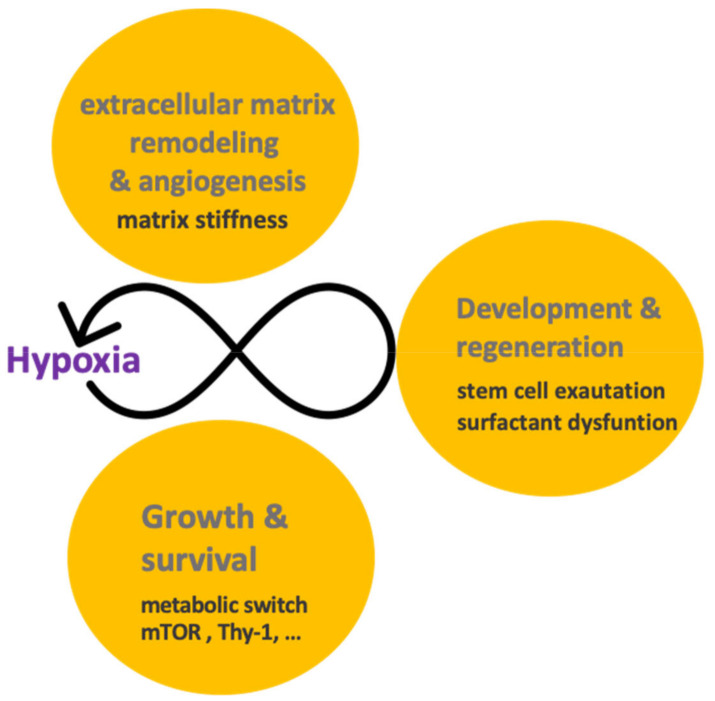
Persistent hypoxia activation. In cancer and fibrosis, hypoxia is a key participant in several processes that have feedback loops, especially when these processes have dysfunctions that promote hypoxia perpetuation (in black).

**Table 1 ijms-22-08335-t001:** Cancer hallmarks associated with hypoxia.

Hallmark	Cause or Effect	Associated with Fibrosis	References
Tumor microenvironment	Cause	---	[28,52,53,54,60,65,66,68,82,86,87]
Metabolic switch	Effect	+	[30,31,42,43,44,45,51,59,62]
Cyclic hypoxia	Cause	+	[32,33,34,35,36,37,38,39]
Cancer stem cell	Effect	+	[34,35,36]
Cancer-associated fibroblasts	Effect	+++	[67,82,83,84]
ECM remodeling	Effect/Cause	+++	[55,67,83]

**Table 2 ijms-22-08335-t002:** Fibrosis hallmarks associated with hypoxia.

Hallmark	Cause or Effect	Associated with Cancer	References
Fibroblast foci	Cause	---	[114,115,117,120,129,130]
Metabolic switch	Effect	+++	[129,131,132,133,134]
Cyclic hypoxia	Cause	+	[124,125]
Fibroblast activation/differentiation	Effect	+	[115,117,120,129,131,132]
Stem cell	Effect	++	[135,136,137,138,139]
ECM remodeling	Effect/Cause	+	[115,117,120]

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
