# Peer review of "Hypoxia in Cancer and Fibrosis: Part of the Problem and Part of the Solution"

_ijms, 2021, doi:10.3390/ijms22158335_

Round 1
Reviewer 1 Report
In this review Romero and Galvez describe the link between Hypoxia pulmonary fibrosis and cancer. It is well-written and covers the topic in a comprehensive way. The importance of hypoxia and HIF signaling in cancer development (including lung cancer) and its link with pulmonary fibrosis this review can be very helpful and informative for the potential readers.
However there are some and subjects that must be included prior to acceptance.
In the first section of the review describing HIF-α domain function there is also an NES domain described in literature (Mylonis et al, 2008) and HIF-1α import is also regulated by importins 4/7 (N-terminal part of HIF-α) (Chachami et a 2009).
In the second section of the review although it mentions signaling events like mTOR it lacks mention to major pathways like EGFR signaling that are characteristics of Hypoxia (Shimoda and Semenza 2011; Kozlova et al 2019), Pulmonary cancer (e.g. Kobayashi et al, 2005; D. Li et al 2007) and Fibrosis (e.g. Shochet et al 2019) and discuss the possibility of linking these phenomena. In the same section although the authors mention HIF-α oxygen independent expression they should mention VHL deficiencies that are causing HIF deregulation and carcinogenesis.
Concerning the figures, in the second figure grey letters in yellow background are difficult to read (please alter). The authors could also add an other figure that comprehensively summarizes common pathways that link Hypoxia, fibrosis and cancer.
Finally, in the last section the authors should expand a little more the paragraph on the therapeutic potential of targeting hypoxia.
Author Response
Response to Reviewer 1 Comments
In this review Romero and Galvez describe the link between Hypoxia pulmonary fibrosis and cancer. It is well-written and covers the topic in a comprehensive way. The importance of hypoxia and HIF signaling in cancer development (including lung cancer) and its link with pulmonary fibrosis this review can be very helpful and informative for the potential readers.
However there are some and subjects that must be included prior to acceptance.
Point 1: In the first section of the review describing HIF-α domain function there is also an NES domain described in literature (Mylonis et al, 2008) and HIF-1α import is also regulated by importins 4/7 (N-terminal part of HIF-α) (Chachami et a 2009).
Reponse 1. As suggested, we included in the section this paragraph:
Furthermore, an additional interaction site has been reported with importins 4 and 7, which is more efficient than NLS interaction [21]. f. The nuclear export signal (NES), located in amino acids 616–658, has been reported to be regulated by MAPK phosphorylation [22].
In the second section of the review although it mentions signaling events like mTOR it lacks mention to major pathways like EGFR signaling that are characteristics of Hypoxia (Shimoda and Semenza 2011; Kozlova et al 2019), Pulmonary cancer (e.g. Kobayashi et al, 2005; D. Li et al 2007) and Fibrosis (e.g. Shochet et al 2019) and discuss the possibility of linking these phenomena. In the same section although the authors mention HIF-α oxygen independent expression they should mention VHL deficiencies that are causing HIF deregulation and carcinogenesis.
Reponse 2. Thank you, this valuable information was already included in the paper.
Concerning the figures, in the second figure grey letters in yellow background are difficult to read (please alter). The authors could also add an other figure that comprehensively summarizes common pathways that link Hypoxia, fibrosis and cancer.
Reponse 3. In the new version we add additional figure as recommended. Thanks.
Finally, in the last section the authors should expand a little more the paragraph on the therapeutic potential of targeting hypoxia.
Reponse 4. Following your suggestions, we are now adding a paragraph about therapeutics.
Reviewer 2 Report
This Review desperately needs accurate revision by an English-speaking peer. Many terms are obsolete or misleading, and many sentences are without verb. I am sure that the manuscript would utmost benefit from better English because using more accurate terms and revising several sentences would greatly clarify the message, that is now quite obscure and difficult to grasp.
I think that the content may be of great interest, but it must be better organised by using more headers and some sub-headers. As a matter of facts, navigation through the text is made difficult not only by inappropriate language, but also by unclear logic to connect the various parts. The Authors are urged to rewrite their manuscript starting from a more stringent logic, and to accompany the text with appropriate schemes or graphs. Unfortunately, Figures 1 and 2 are not informative at all, and the tables that list the various hallmarks without distinguishing causes and effects need more insight to be useful.
Finally, the Authors are urged to define aims and rationale, with more focused conclusions that now are general and inconclusive.
Author Response
Response to Reviewer 2 Comments
This Review desperately needs accurate revision by an English-speaking peer. Many terms are obsolete or misleading, and many sentences are without verb. I am sure that the manuscript would utmost benefit from better English because using more accurate terms and revising several sentences would greatly clarify the message, that is now quite obscure and difficult to grasp.
Reponse 1. We apologize for not being clear. As suggested, additional revision was done with a native speaker.
I think that the content may be of great interest, but it must be better organised by using more headers and some sub-headers. As a matter of facts, navigation through the text is made difficult not only by inappropriate language, but also by unclear logic to connect the various parts. The Authors are urged to rewrite their manuscript starting from a more stringent logic, and to accompany the text with appropriate schemes or graphs. Unfortunately, Figures 1 and 2 are not informative at all, and the tables that list the various hallmarks without distinguishing causes and effects need more insight to be useful.
Reponse 2. In this new version the organization of the article was completely changed including a new figure and more headers and some sub-headers as recommended.
Finally, the Authors are urged to define aims and rationale, with more focused conclusions that now are general and inconclusive.
Reponse 3. A paragraph concerning this was added in the conclusions and abstract in the revised manuscript. Thanks
Round 2
Reviewer 2 Report
The manuscript has improved, and now one can focus on scientific issues (but how much time could have been saved if this was done on the first submission!). Nevertheless, there are still many other syntax/grammar and scientific issues that detract from its suitability to be published in the present form. Some of them are hand-marked in the attached pdf, here I highlight the most important points.
The lung is, among the body organs, the one that is exposed to the highest PO2. Moreover, lung cells, of which the alveolar epithelial cells constitute a major part, are oxygenated directly by atmospheric air, not by arterial blood as other organs, thereby eluding, at least in part, the issues related to the proliferation of the vascular network. One can therefore expect that, although the molecular mechanisms that sense oxygen are the same, the issues related to the oxygen sensitivity (e.g., the KM for oxygen of the HIF hydroxylation reaction) and perhaps to the molecular responses to hypoxia may become preponderant. Thus, to address the role of HIF in pulmonary tissues, the Authors are urged to discuss the issues related to the HIF machinery in the lungs.
Figure 1 is scarcely informative. It is not O2 that becomes OH, but an amino acid in HIF-1 alpha becomes hydroxylated due to the presence of O2.
Often the Authors mix together information related to lung cancer with that related to other cancer types of, but this should be kept to a minimum and they must clearly state this.
The issues related to the redox imbalance (or oxidative stress) are quite relevant. First, ROS sometimes contribute to engage HIF stabilization, and the major component of OSAS may not be the (short) periods of hypoxia but rather the redox imbalance, However, this pivotal issue is not discussed here.
Likewise, the Authors completely neglect the immunological modulation, which may be a critical connection between hypoxia and fibrosis.

Author Response
The manuscript has improved, and now one can focus on scientific issues (but how much time could have been saved if this was done on the first submission!). Nevertheless, there are still many other syntax/grammar and scientific issues that detract from its suitability to be published in the present form. Some of them are hand-marked in the attached pdf, here I highlight the most important points.
Point 1. The lung is, among the body organs, the one that is exposed to the highest PO2. Moreover, lung cells, of which the alveolar epithelial cells constitute a major part, are oxygenated directly by atmospheric air, not by arterial blood as other organs, thereby eluding, at least in part, the issues related to the proliferation of the vascular network. One can therefore expect that, although the molecular mechanisms that sense oxygen are the same, the issues related to the oxygen sensitivity (e.g., the KM for oxygen of the HIF hydroxylation reaction) and perhaps to the molecular responses to hypoxia may become preponderant. Thus, to address the role of HIF in pulmonary tissues, the Authors are urged to discuss the issues related to the HIF machinery in the lungs.
Response 1. This is a relevant point. We consider this to suggest that the role of hypoxia is related to progression rather than disease onset or lung oxygenation. Therefore, we refer that hypoxia occurs in solid tumors and fibroblast foci already established. We add the following paragraph:
Lung cancer and idiopathic pulmonary fibrosis (IPF) compromise the lung parenchyma with an irreversible loss of gas exchange, solid tumors and fibroblast foci have limited nutrient supply due to a dense microenvironment, counterintuitively, cells still proliferate causing disease progression. In this context, hypoxia adaptation could represent a coherent link between cell proliferation and interruption of nutrients supply.
Point 2. Figure 1 is scarcely informative. It is not O2 that becomes OH, but an amino acid in HIF-1 alpha becomes hydroxylated due to the presence of O2.
Response 2. Thanks for this observation. The new figure incorporates these changes.
Point 3. Often the Authors mix together information related to lung cancer with that related to other cancer types of, but this should be kept to a minimum and they must clearly state this.
Response 3. In this new version, we indicate when information is not related to lung cancer. Also, we considered that this interconnection between cancer and fibrosis is not exclusively in the lungs.
Point 4. The issues related to the redox imbalance (or oxidative stress) are quite relevant. First, ROS sometimes contribute to engage HIF stabilization, and the major component of OSAS may not be the (short) periods of hypoxia but rather the redox imbalance, However, this pivotal issue is not discussed here.
Response 4. As mentioned, in this revised version we are including this information in the text.
Point 5. Likewise, the Authors completely neglect the immunological modulation, which may be a critical connection between hypoxia and fibrosis.
Response 5. Following this recommendation, we add the following paragraph:
Additionally, the role of other cell populations (in addition to epithelial cells and fibroblasts) has been little studied in IPF. However, it has been reported a population of macrophages that can promote fibrosis through immunological modulation [104]. During early events of fibrosis, macrophages have been implicated in the activation of host defense and profibrotic mechanisms [105]. Furthermore, the hypoxic microenvironment has an altered metabolite composition that promotes the activation of immune cells [106]. Accordingly, the link might be through elevated mitochondrial ROS or TCA cycle metabolites already involve in the regulation of immune cells [106,107].
Round 3
Reviewer 2 Report
The Authors have answered satisfactorily to my concerns. Although there is still a great margin of improvement, this manuscript is now suitable to be published in the present form. English, however, requires a further revision.